# Resveratrol Promotes Tumor Microvessel Growth via Endoglin and Extracellular Signal-Regulated Kinase Signaling Pathway and Enhances the Anticancer Efficacy of Gemcitabine against Lung Cancer

**DOI:** 10.3390/cancers12040974

**Published:** 2020-04-15

**Authors:** San-Hai Qin, Andy T. Y. Lau, Zhan-Ling Liang, Heng Wee Tan, Yan-Chen Ji, Qiu-Hua Zhong, Xiao-Yun Zhao, Yan-Ming Xu

**Affiliations:** Laboratory of Cancer Biology and Epigenetics, Department of Cell Biology and Genetics, Shantou University Medical College, Shantou, Guangdong 515041, China; sanhaiqin@stu.edu.cn (S.-H.Q.); andytylau@stu.edu.cn (A.T.Y.L.); 16zlliang@stu.edu.cn (Z.-L.L.); hwtan@stu.edu.cn (H.W.T.); joyceycji@stu.edu.cn (Y.-C.J.); 17qhzhong@stu.edu.cn (Q.-H.Z.); 18xyzhao@stu.edu.cn (X.-Y.Z.)

**Keywords:** resveratrol, gemcitabine, endoglin, ERKs, microvessel growth, lung cancer

## Abstract

The synergistic anticancer effect of gemcitabine (GEM) and resveratrol (RSVL) has been noted in certain cancer types. However, whether the same phenomenon would occur in lung cancer is unclear. Here, we uncovered the molecular mechanism by which RSVL enhances the anticancer effect of GEM against lung cancer cells both in vitro and in vivo. We established human lung adenocarcinoma HCC827 xenografts in nude mice and treated them with GEM and RSVL to detect their synergistic effect in vivo. Tumor tissue sections from nude mice were subjected to hematoxylin and eosin staining for blood vessel morphological observation, and immunohistochemistry was conducted to detect CD31-positive staining blood vessels. We also established the HCC827-human umbilical vein endothelial cell (HUVEC) co-culture model to observe the tubule network formation. Human angiogenesis antibody array was used to screen the angiogenesis-related proteins in RSVL-treated HCC827. RSVL suppressed the expression of endoglin (ENG) and increased tumor microvessel growth and blood perfusion into tumor. Co-treatment of RSVL and GEM led to more tumor growth suppression than treatment of GEM alone. Mechanistically, using the HCC827-HUVEC co-culture model, we showed that RSVL-suppressed ENG expression was accompanied with augmented levels of phosphorylated extracellular signal-regulated kinase (ERK) 1/2 and increased tubule network formation, which may explain why RSVL promoted tumor microvessel growth in vivo. RSVL promoted tumor microvessel growth via ENG and ERK and enhanced the anticancer efficacy of GEM. Our results suggest that intake of RSVL may be beneficial during lung cancer chemotherapy.

## 1. Introduction

Lung cancer is one of the worldwide malignancies with the highest incidence and mortality, which has been ranked number one in terms of morbidity and accounts for 11.6% of the total incidence of cancers [1]. Gemcitabine (GEM), a first-line chemotherapeutic agent for advanced non-small cell lung cancer (NSCLC), is commonly used but not very effective as a single agent, and therefore it is important to find ways to enhance its therapeutic efficacy.

Angiogenesis is crucial during the development of human lung cancer, in which cancer cells secrete angiogenic factors and induce neovascularization to establish tumor vascular network, providing nutrients required for cell expansion, facilitating their growth, proliferation, and metastasis. Traditionally, anti-angiogenic therapy should be a very logical strategy for lung cancer therapy [2]. However, it should also be noted that because the tumor vasculatures are abnormal and consist of chaotic labyrinth of malformed and destabilized vessels that are structurally and functionally impaired, the tumor microenvironment is therefore not only hypoxic and acidic but also is surrounded by high interstitial pressure, which acts as a pathologic barrier for drug delivery into tumors and leads to a notably reduced therapeutic effect [3]. Traditional anti-angiogenic approaches often cause extreme hypoxia in tumors and eventually lead to increased drug resistance, local invasion, and more distant metastasis [4]. In addition, angiogenesis compensatory pathways and alternative modes of tumor vascularization, such as vascular co-option or mimicry, may also play roles in resistance to anti-angiogenesis therapy [5]. Suffice to say, the clinical efficacies of current angiogenesis inhibitors are limited and some of them are totally invalid or unacceptably toxic [6], and therefore new, safer, and more effective agents are urgently needed.

Resveratrol (trans-3,5,4′-trihydroxystilbene, RSVL) is a well-known plant-derived natural polyphenolic compound that widely presents in grapes, berries, peanuts, and is abundant in red wine, exerting extensive bioactivities including antioxidative, anticancer, antiaging, anti-inflammatory, and other effects [7,8,9,10,11]. However, opposing results were obtained among studies that showed that RSVL possesses both anti- and pro-angiogenesis effects, depending on model systems and circumstances [12]. Combination of RSVL with chemotherapeutic drugs was found to enhance the efficacy of these drugs, such as the fact that (1) RSVL sensitizes pancreatic cancer MIA-PaCa-2 cells to chemotherapeutic agents such as docetaxel, mitoxantrone, 5-fluorouracil, cisplatin, and oxaliplatin [13]; (2) RSVL decreases Rad51 expression and sensitizes cisplatin-resistant MCF-7 breast cancer cells [14]; (3) RSVL markedly potentiates the effect of sorafenib in hepatocellular carcinoma Hep3b cells [15]; (4) RSVL enhances the apoptotic and oxidant effects of paclitaxel in DBTRG glioblastoma cells [16]; and (5) RSVL sensitizes colorectal cancer HCT116 and HT-29 cells to doxorubicin [17]. However, whether RSVL can potentiate the effect of GEM in lung cancer is unclear.

In this study, we investigated whether RSVL could enhance the anticancer effect of GEM against human lung cancer cells both in vitro and in vivo. We established a HCC827-human umbilical vein endothelial cell (HUVEC) co-culture model and examined the effect of RSVL on tubule network formation in vitro. We also performed hematoxylin and eosin (HE) staining for blood vessel morphology observation and immunohistochemistry (IHC) to detect CD31-positive staining blood vessels in tumor tissue sections of nude mice with HCC827 xenografts, which were conducted to examine the effect of RSVL in tumor microvessel growth in vivo. Mechanistically, the present work showed that the downregulated protein expression level of endoglin (ENG) and the activation of extracellular signal-regulated kinase (ERK) signaling pathway play important roles in RSVL-promoted tumor microvessel growth, leading to increased blood perfusion and drug delivery into tumor and thereby resulting in enhanced anticancer effect of GEM. The implications of our findings suggest the potential clinical applications of RSVL to enhance the anticancer efficacy of anticancer drugs against lung cancer.

## 2. Results

### 2.1. RSVL in Combination with GEM Showed No Synergistic Effects on HCC827 Cancer Cells Cultured In Vitro

To determine whether RSVL, a stilbene, might have a role in the treatment of lung cancer in combination with GEM, we firstly investigated the effect of RSVL on the proliferation of HCC827 lung cancer cell line to determine a suitable concentration. We undertook BEAS-2B bronchial epithelial cells and human umbilical vein endothelial cells (HUVEC) as normal controls. Upon treatment of these cells with RSVL, it can be seen that RSVL at the concentration of 5–10 µM showed negligible cytotoxicity to HCC827 lung cancer cells, BEAS-2B, and HUVEC at 24 h (Figure 1A). Because the concentration at 10 µM was relatively low and non-cytotoxic, and more importantly, RSVL at 10 µM was water-soluble and may have reached the indicated concentration in vivo, thus RSVL was used in most of the subsequent experiments at the concentration of 10 µM. Next, we investigated whether RSVL can potentiate the effect of GEM against HCC827 lung cancer cells cultured in vitro, however, there was no significant difference in cell viability between GEM treated alone and GEM combined with RSVL on HCC827 cells (Figure 1B).

### 2.2. RSVL Enhanced the Anticancer Efficacy of GEM in HCC827 Lung Cancer Bearing Nude Mice

From the information above, we can see that there was no observable synergistic effect of RSVL in GEM-treated HCC827 cancer cell culture in vitro. However, we wondered whether this was due to the simplicity of the experimental design (only a monolayer of cancer cell culture in a 24-well plate) because in reality the tumor microenvironment is so complex and many cell–cell interactions are actually involved. For this reason, we examined the therapeutic potential of RSVL and GEM either alone or in combination on the growth of transplanted HCC827 human lung cancer cells in nude mice. The experimental protocol is depicted in Figure 2A. Briefly, HCC827 cells were subcutaneously inoculated into the right flanks of nude mice. After 7 days, we randomized the animals into four groups and started the treatment following the experimental protocol. Tumors were measured twice a week, and after administration of 25 days, mice were sacrificed and tumors were excised surgically and weighed, and then were fixed in 4% formaldehyde solution for further study. Compared with GEM treated alone, the combination of the two agents was more effective in reducing the tumor burden. The tumors in the group of combination grew slower, appearing with lower volume and weight, as well as a lower tumor growth rate (Figure 2B–E). These results showed that RSVL enhanced the anticancer efficacy of GEM against HCC827 lung cancer in vivo in xenograft-bearing nude mice.

### 2.3. RSVL Increased Microvessel Growth and Promoted Blood Perfusion into Tumor in Lung Cancer Xenograft Mice

From the above results, it is quite intriguing that RSVL enhanced the anticancer efficacy of GEM against HCC827 lung cancer in vivo but not in vitro. To answer this question, we made tumor tissues from nude mice into sections and performed HE staining for the morphology observation. The results showed that there were more tumor microvessels and bloodstream in RSVL or combined treatment groups as compared with control or groups treated with GEM alone (Figure 3A,B). The results of immunohistochemistry (IHC) assay also indicated increased CD31-positive staining blood vessels in RSVL or combined treatment groups (Figure 3C,D), suggesting that RSVL increases tumor microvessel growth and promotes blood perfusion into tumor in lung cancer-transplanted nude mice.

### 2.4. RSVL Promoted Tubule Network Formation in HCC827-HUVEC Co-Culture Model

To confirm our findings in vivo, we used enhanced green fluorescent protein (EGFP) stably-expressing HUVEC that was cultured with HCC827 cells to establish the HCC827-HUVEC co-culture model, and observed tubule network formations in vitro under the fluorescent microscope after RSVL treatment. At the end, photos were captured and analyzed. Experimental protocol is depicted in Figure 4A. After treatment with 10 µM RSVL for 24 h, the HCC827-HUVEC co-culture model showed better tubular formation, appearing to have a higher percentage of elongation, tubules, and junctions formed, whereas HUVEC alone/BEAS-2B-HUVEC co-culture model showed no significant difference in tubule network formation (Figure 4B,C). The above evidence suggests that RSVL enhanced the anticancer efficacy of GEM against lung cancer in vivo, which may be explained by the promoted microvessel growth and blood perfusion in tumor, increasing the concentration of GEM in the surrounding interstitial space, thereby enhancing its anticancer efficacy against lung cancer.

### 2.5. RSVL Suppressed both the mRNA and Protein Levels of ENG in HCC827 Lung Cancer Cells, and also Decreased the Protein Level of ENG in Tumor Tissues from HCC827 Xenograft Mice

Without any clues on which angiogenic factors might possibly be involved in the above phenomenon, we resolved to screen the differential expression of 55 angiogenesis-related protein targets in HCC827 cancer cells after RSVL treatment by using the human angiogenesis array. According to the screening results, one of the downregulated targets, endoglin (ENG), caught our attention (Figure 5). We performed qPCR and Western blotting again, finding that RSVL suppressed both mRNA and protein levels of ENG in HCC827 cells cultured in vitro (Figure 6A,B), which is consistent with the angiogenesis array screening result. Next, we conducted IHC to detect the protein levels of ENG in tumor tissue sections from HCC827 xenograft-bearing nude mice. Notably, ENG-positive staining became weaker in RSVL alone and RSVL + GEM groups as compared with control or groups treated with GEM alone (Figure 6C,D), which suggested that RSVL also decreased the protein level of ENG in vivo.

### 2.6. ENG Was Crucial in RSVL-Promoted Microvessel Growth

As indicated in previous sections, ENG expressions were suppressed by RSVL both in vitro and in vivo, which suggested ENG may play an important role in RSVL-promoted microvessel growth. Thus, we focused on the study of ENG.

First, we transfected ENG-small interfering RNA (siRNA) into HCC827 cells to knockdown the endogenous ENG, then mixed them with HUVEC to establish the HCC827-HUVEC co-culture model and observed the tubule network formation under the fluorescent microscope. The knockdown efficiency of ENG- or control (CTRL)-siRNA was validated (Appendix A). After knockdown of ENG, HUVEC showed better tubular formation, appearing to have a higher percentage of elongation, tubules, and junctions formed as compared with CTRL-siRNA group (Figure 7A,B), which indicated the fact that knockdown of ENG can promote tubule network formation in HCC827-HUVEC co-culture model. We also constructed the plasmid of pcDNA3.1(+)-ENG-mCherry and transfected it into HCC827 cells to overexpress ENG, then cultured them with HUVEC to establish the HCC827-HUVEC co-culture model and observed the tubule network formations under fluorescence microscope. However, after overexpression of ENG, HUVEC showed no differences in tubule network formations compared with the control vector group both in the presence or absence of RSVL for 24 h (Figure 7C–E), suggesting that overexpression of ENG can inhibit tubule network formation induced by RSVL in HCC827-HUVEC co-culture model. Thus, it can be seen that ENG plays a crucial role in RSVL-promoted tumor microvessel growth, of which the expression levels of ENG are negatively correlated with tubule network formations in the HCC827-HUVEC co-culture model.

### 2.7. ENG and ERK Signaling Pathway Played Important Roles in RSVL-Promoted Tumor Microvessel Growth

Some reports demonstrated that the ERK signaling pathway is involved in endothelial cell growth and migration [18,19]. To investigate the roles of ERK signaling pathway in RSVL-promoted tumor microvessel growth, we performed Western blotting to detect the levels of phosphorylated ERK1/2 in HCC827-HUVEC co-culture model.

First, we treated HCC827 cells with RSVL and found that levels of p-ERK1/2 decreased in a dose-dependent manner from 10 to 50 µM (Figure 8A). Then, we manipulated the ENG protein levels by knockdown or overexpression of ENG in HCC827 cells co-cultured with HUVEC, and found that the levels of p-ERK1/2 increased after knockdown of ENG (Figure 8B, left panel) whereas they decreased after overexpression of ENG (Figure 8C, left panel) in the HCC827-HUVEC co-culture models. Of note, the increase of the levels of p-ERK1/2 was likely contributed to by the HUVEC in the co-culture model as we can see that the level of p-ERK1/2 was increased in the co-culture situation (Figure 8B, left panel), but decreased in HCC827 alone (Figure 8B, right panel) (which further supports our data in Figure 5 and Figure 8A showing that downregulation of ENG by RSVL caused decrease of p-ERK1/2 level in HCC827 cells). Therefore, the ENG expression level in HCC827 cells corresponded with the degree of ERK1/2 activation in HUVEC in the HCC827-HUVEC co-culture model.

Furthermore, we also blocked the phosphorylation of ERKs by treatment with MEK inhibitor PD0325901. We found that HUVEC showed obvious tubular formations stimulated by RSVL treatment, whereas HUVEC showed impaired tubule network formations after PD0325901 treatment, no matter whether RSVL was present or not (Figure 8D,E). The above results indicated that the ENG expression level negatively corresponded with the degree of ERK1/2 activation in the HCC827-HUVEC co-culture model. Treatment of RSVL reduced the ENG expression and led to the attenuation of ERK phosphorylation inhibition, resulting in increased tumor microvessel growth, which can be suppressed by PD0325901. Collectively, the ERK signaling pathway may play an important role in RSVL-promoted tumor microvessel growth.

### 2.8. Data from Online Databases Suggest Increased Expression of ENG May Be Negatively Correlated to the Survival of Lung Cancer Patients

Lastly, to correlate our findings more clinically, we analyzed the ENG mRNA levels in patients from an online database (The Cancer Genome Atlas (TCGA) Provisional 2015) and found that a high percentage (3.72%–10.24%) of lung cancer patients had altered ENG gene expression, and in most cases, ENG was up-regulated instead of down-regulated. There were more lung cancer patients with adenocarcinoma that showed increased ENG expression than patients with squamous cell carcinoma (Figure 9A). Increased expression of ENG was negatively-correlated with the survival of lung cancer patients. However, when we performed the survival analysis based on different cancer histology, we found that the ENG expression specifically had a strong impact on the survival of patients with adenocarcinoma but not those with squamous cell carcinoma (Figure 9B). These results indicated that a high level of ENG may be an elevated risk factor in the development of lung cancer, and thus decreased ENG induced by RSVL will be favorable for cancer prevention, especially for lung adenocarcinoma prevention.

## 3. Discussion

In recent years, many natural products have been recognized as anticancer agents, and previous studies have indicated the synergistic anticancer effect of GEM and RSVL in certain cancer types, such as pancreatic cancer and ovarian carcinoma [20,21]. However, whether the same phenomenon would occur in lung cancer is unclear. Here, we delineated for the very first time the molecular mechanism by which RSVL enhances chemosensitivity and the critical role of ENG in GEM-treated human lung adenocarcinoma cell line HCC827.

ENG (also known as CD105) is a cell membrane glycoprotein mainly expressed in endothelial cells and that is overexpressed in tumor-associated vascular endothelium, which functions as an accessory component of the transforming growth factor-β (TGF-β) receptor complex and is involved in vascular development and remodeling. In solid malignancies, ENG is almost exclusively expressed in endothelial cells of both peri- and intra-tumoral blood vessels and on tumor stromal components. Several studies have defined the role of ENG as a powerful marker to quantify intratumoral microvessel density (IMVD) in solid and hematopoietic tumors, including breast, prostate, cervical, colorectal, and non-small cell lung cancer (NSCLC), and in multiple myeloma and hairy cell leukemia. Quantification of tumor microvessel density, as determined by immunohistochemical staining for ENG, is a significant indicator of poor prognosis in patients with selected solid neoplasias including NSCLC, cervical cancer, prostate cancer, and breast carcinoma [22]. Data from online databases suggest that increased expression of ENG is negatively correlated with the survival of lung cancer patients. In this case, the fact that RSVL decreased ENG level *in vitro* and *in vivo* in our model systems here may suggest the beneficial therapeutic role of RSVL in conjunction with chemotherapy (such as GEM) for lung cancers.

Lee and Blobe [23] reported that β-arrestin2 binding to ENG causes the internalization of ENG and simultaneous accumulation of ENG and β-arrestin2 in endocytic vesicles, which antagonized TGF-β-mediated ERK signaling, altered the subcellular distribution of activated ERK, and inhibited endothelial cell migration in a manner dependent on the ability of ENG to interact with β-arrestin2. Moreover, ENG impedes endothelial cell growth through sustained inhibition of ERK-induced c-Myc and cyclinD1 expression in a TGF-β-independent manner, by which ENG augments growth-inhibition by targeting ERK and key downstream mitogenic substrates [19]. In our current study, as summarized in Figure 10, RSVL inhibited the expression of ENG in HCC827 lung cancer cells, corresponding with decreased ENG in the surrounding interstitial space and microvessels in tumor tissues by direct physical contact or in a paracrine manner. Subsequently, the ERK signaling towards endothelial growth and migration was activated, which contributed to enhanced tubule network formation and microvessel growth. Downregulation of ENG plays a crucial role in RSVL-promoted tumor microvessel growth, which leads to increased blood perfusion and drug delivery into tumor, thereby resulting in an enhanced anticancer effect of GEM.

The participation of cancer cells, as well as growth factors released by tumor cells and endothelium–extracellular matrix (ECM) interactions are highlighted in tumor angiogenesis, as well as the physical contacts and the paracrine actions that are the keys to endothelial cell (EC) differentiation [24,25]. Many cancer cell lines, such as hepatocellular carcinoma HepG2 and human neuroblastoma SK-N-SH, possess the ability to induce EC morphological changes, whereas for normal cells, such as human embryonic kidney HEK-293, liver cell L0-2, human fiber cell IMR90, and human smooth muscle cell VSMC, also possess the ability to induce EC morphological changes. Compared to Matrigel model, a common model for studying tubule network formation in vitro, the co-culture model simulates the real angiogenic microenvironment in the human body, which allows direct interactions between cancer cells and ECs, thus facilitating the study of factors and signaling pathways governing blood vessel formation in cancer [26,27]. In this study, we used the HCC827-HUVEC co-culture model to evaluate the effect of RSVL on tumor angiogenesis and its possible mechanism.

The concentrations of RSVL in most anticancer studies are often beyond 20 µM and above, some of them even reach 100 µM [28,29,30,31,32]. However, due to the low water solubility and bioavailability of RSVL, plasma levels as high as above 20 µM may not be physiologically attainable in humans [33]. Moreover, the high concentration of RSVL may be cytotoxic to normal cells. In contrast, the concentration we used in our experiment (10 µM) is relatively low and non-cytotoxic to BEAS-2B, HUVEC, and HCC827 lung cancer cell lines, and more importantly, RSVL at 10 µM is water-soluble and may reach the indicated concentration in serum and even achieve higher levels of drug accumulation in tumor tissues, which in turn will inform the need for dietary advice on the intake of RSVL for the patients undergoing chemotherapy. Nevertheless, our findings indicate that RSVL may have the potential to augment the therapeutic efficacy of anticancer drugs and suggest that consumption of RSVL may be beneficial during cancer therapy. Further study of RSVL in combination of other well-known anticancer drugs (besides GEM) is warranted, for attesting whether similar phenomenon would occur.

## 4. Materials and Methods

### 4.1. Reagents

RSVL (98%) was purchased from Sigma-Aldrich (St. Louis, MO, USA). GEM and PD0325901 were purchased from Selleck Chemicals (Shanghai, China). All other general chemicals were purchased from Sigma-Aldrich and GE Healthcare (Uppsala, Sweden). Antibodies used for immunoblotting and immunohistochemical staining were purchased from Santa Cruz Biotechnology (Santa Cruz, CA, USA) and Cell Signaling Technology (Danvers, MA, USA).

### 4.2. Cell Lines and Culture Conditions

All cell lines used were purchased from the American Type Culture Collection (ATCC) (Rockville, MD). The human lung adenocarcinoma cell line HCC827 (CRL-2868) harbors an acquired mutation in the EGFR tyrosine kinase domain (exon 19, del E746-A750), and this mutation has been verified through gene sequencing. Cells were routinely grown in RPMI-1640 complete medium containing 10% FBS, 100 U/mL penicillin, and 100 μg/mL streptomycin at 37 °C in an atmosphere of 5% CO_2_/95% air, as recommended by ATCC. The normal human bronchial epithelial cell (BEAS-2B) and human umbilical vein endothelial cell (HUVEC) were cultured in standard culture conditions, as recommended by the ATCC.

### 4.3. Plasmids, Small Interfering RNAs (siRNAs), and Transfection

The pEGFP-N1 plasmid was from Clontech (Mountain View, CA). The pcDNA3.1(+)-ENG-mCherry encoding plasmid and siRNAs were synthesized by IGE Biotechnology Ltd. (Guangzhou, China). Cells were transfected with the above expression plasmids, with either siRNA duplexes against ENG- (5′-AGAAAGAGCUUGUUGCGCA-3′) or CTRL-siRNA (a scrambled sequence that will not lead to the specific degradation of any known cellular mRNA) [34], using LipofectAMINE 2000 (Invitrogen, Carlsbad, CA).

### 4.4. Cell Proliferation Assay

The effects of RSVL on proliferation of cells were determined by naphthol blue black (NBB) staining assay. Briefly, HCC827, BEAS-2B, or HUVEC cells (5 × 10^4^ per well) were seeded in triplicate in a 24-well plate. After culturing for 24 h, the cells were treated with various concentrations of RSVL and were further incubated for 24 h; cells were then fixed with 10% formalin for 8 min and stained with 0.05% NBB solution for 30 min, the wells were washed by distilled water for three times, 50 mM NaOH was added to each well, and the absorbance of the cell suspension was measured at 595 nm using a 96-well multiscanner (Thermo Scientific Multiskan FC, Thermo Scientific, USA).

### 4.5. HCC827-HUVEC Co-Culture Model Establishment and Image Analysis

After transfection with pEGFP-N1 and antibiotic screening by G418, HUVEC with stable EGFP expression were constructed and cultured with HCC827 cells to establish the HCC827-HUVEC co-culture model following the method described previously [27]. Briefly, HCC827 and HUVEC-EGFP cells were harvested by trypsin digestion and mixed in ratios of 1:1 before seeding them in 12-well plates (3 × 10^5^ per well). BEAS-2B cells co-cultured with HUVEC-EGFP cells or HUVEC-EGFP cells cultured alone were served as controls. There were three replicates in each group, and after treatment with RSVL, images were captured under a ZEISS Observer A1 inverted fluorescence microscope (ZEISS, Germany) and analyzed with Angiogenesis Analyzer software with ImageJ plugin. Quantification of tubule network formation was obtained by averaging the number of junctions, number of tubules, and total branching lengths.

### 4.6. In Vivo Xenograft Mouse Model Establishment and Treatment

Athymic nude mice were purchased from Beijing Vital River Laboratory Animal Technology Co., Ltd. Animals were maintained under “specific pathogen-free” conditions and had free access to food and water. All animal studies were conducted according to guidelines approved by the Shantou University Medical College Institutional Animal Care and Use Committee. The ethic code is SUMC2019-321. HCC827 cells (2 × 10^6^ in 200 μL RPMI) were injected s.c. into the right flank of mouse, and tumor growth was monitored. Approximately 1 week after tumor cell inoculation, when the HCC827 xenografts were growing to suitable size, mice were randomly assigned to four groups (*n* = 5) and treated with vehicle, RSVL (1 µmol kg^−1^, five times weekly by i.v. injection), GEM (25 mg kg^−1^, twice weekly by i.p. injection), or both. Mice were weighed and tumors were measured by caliper twice weekly. Tumor volume was calculated according to the following formula: tumor volume (mm^3^) = 0.5 × length × width^2^. After administration of 25 days, when tumors reached about 1000 mm^3^ total volume, mice were sacrificed, tumors were excised surgically and weighed, and then were fixed in 4% formaldehyde solution for further study.

### 4.7. RNA Isolation and Conditions for Quantitative Real-Time RT-PCR

Total RNA was extracted with Trizol reagent (Life Technology, NY, USA) and was reverse-transcribed into cDNA using the GoScript RT reagent mix (Promega Corporation, Madison, USA) according to the manufacturer’s instructions. For RT-qPCR, 10 ng of cDNA template was amplified by the appropriate primer set in a reaction contained RT2 SYBR Green ROX qPCR Mastermix (Qiagen-SABiosciences, USA). The real-time PCR assay was performed in an ABI QS5 Real-Time PCR Detection System (Applied Biosystems, CA, USA). Primers used in quantitative real-time RT-PCR were designed by using the RTPrimerDB database (www.rtprimerdb.org) [35] or Primer-BLAST (Primer3 and NCBI, Bethesda, USA), and were synthesized by Beijing Genomics Institute (Shenzhen, Guangdong, China; the primer sequences are available upon request). β-actin was used as a reference gene and the relative gene expressions were calculated using the comparative C_T_ method (2^-ΔΔC^_T_ method), as described previously [36].

### 4.8. The Human Angiogenesis Array Screening

The Human Angiogenesis Array Kit (R&D Systems, Minneapolis, MN) was used to detect the relative levels of expression of 55 angiogenesis-related proteins according to the manufacturer’s instructions (The Human Angiogenesis Array coordinates are shown in Appendix A). Briefly, after treatment with RSVL or Dulbecco’s phosphate-buffered saline (as control), HCC827 cells were lysated and the supernatants were collected. The membranes were blocked with Array Buffer in advance, then equal amounts of protein supernatants were incubated with reconstituted Detection Antibody Cocktail and hybridized with diluted Streptavidin-HRP, then Chemi Reagent Mix was added and exposed using a gel imaging and analysis system (Tenan, China). Integrated optical density (IOD) in each spot of the array was analyzed using Gel-Pro analyzer 4, and was compared with corresponding signals on different arrays to determine the relative change in angiogenesis-related proteins.

### 4.9. Western Blotting Analysis

Extracted protein was resolved by 10% SDS-PAGE and transferred to polyvinylidene difluoride membrane. The membrane was then probed with various primary antibodies followed by incubations with appropriate secondary antibodies and subjected to Enhanced Chemiluminescence Western Blotting Detection Reagents (GE Healthcare, PA, USA), as described previously [37]. Antibodies used for Western blot and IHC were purchased from Santa Cruz Biotechnology (Santa Cruz, CA), Cell Signaling Technology (Danvers, MA), and Sigma-Aldrich, with the following dilutions: ENG (sc-20072; Santa Cruz Biotechnology), 1:1000 for Western blot, and 1:100 for IHC; CD31 (sc-13537; Santa Cruz Biotechnology), 1:100 for IHC; p-ERK1/2 Thr^202^/Tyr^204^ (4370; Cell Signaling Technology), 1:1000; ERK1/2 (9102; Cell Signaling Technology), 1:1000; and β-actin (A2228; Sigma-Aldrich), 1:10,000. Whole blot showing all the bands with all molecular weight markers on the Western can be found in Appendix A.

### 4.10. HE Staining and Immunohistochemistry

Tumor tissues from HCC827 xenograft-implanted mice were fixed by formalin and embedded by paraffin, then were made into 4 µm thick sections and stained with hematoxylin and eosin (HE) for histo-morphological evaluation. CD31-positive staining blood vessels as well as ENG protein levels were detected by immunohistochemistry (IHC). CD31 (1:100) and ENG (1:100) were used as primary antibodies. Sections were scanned as digitalized images using Perkin Elmer Mantra Quantitative Pathology Workstation (PerkinElmer, MA, USA). Tumor blood perfusion ratio, which is defined as the ratio of red-dyed blood cell pixel to the total image pixel, was used to quantify blood perfusion in tumor tissue. The number of CD31-positive staining blood vessels was assessed by counting the vascular structures in five high power fields (HPFs, 200× magnification) and then averaging the counts of the five fields. The IHC staining score of each section was determined through assessment, as described previously [38].

### 4.11. Clinical Database and Statistical Analysis

The connections between ENG mRNA expression and lung cancer were investigated using cBioPortal database (www.cbioportal.org) [39]. Datasets of lung cancer patients were obtained from The Cancer Genome Atlas (TCGA) studies. Expressions of ENG with z-score ≥ +1.5 or ≤ −1.5 were considered significantly altered. The z-score is calculated as ((expression in tumor sample – mean expression in normal sample) ÷ standard deviation of expression in normal sample). Survival analysis of lung cancer patients was performed using the Kaplan–Meier plotter database (http://kmplot.com) [40]. SPSS version 23.0 was used for all of the statistical analysis in the study. All quantitative data are expressed as means ± SD, as indicated. Student’s *t*-test or one-way ANOVA was used for statistical analysis. A probability of *p* ≤ 0.05 was used as the criterion for statistical significance.

## 5. Conclusions

In summary, the results from the current study showed that RSVL enhanced the anticancer efficacy of GEM against HCC827 lung cancer in vivo. We discovered a new molecular mechanism in which ENG and ERK signaling pathway played important roles in RSVL-promoted tumor microvessel growth and blood perfusion into tumor, which resulted in enhanced anticancer effect of GEM. Thus, intake of RSVL may be beneficial during lung cancer chemotherapy.

## Figures and Tables

**Figure 1 cancers-12-00974-f001:**
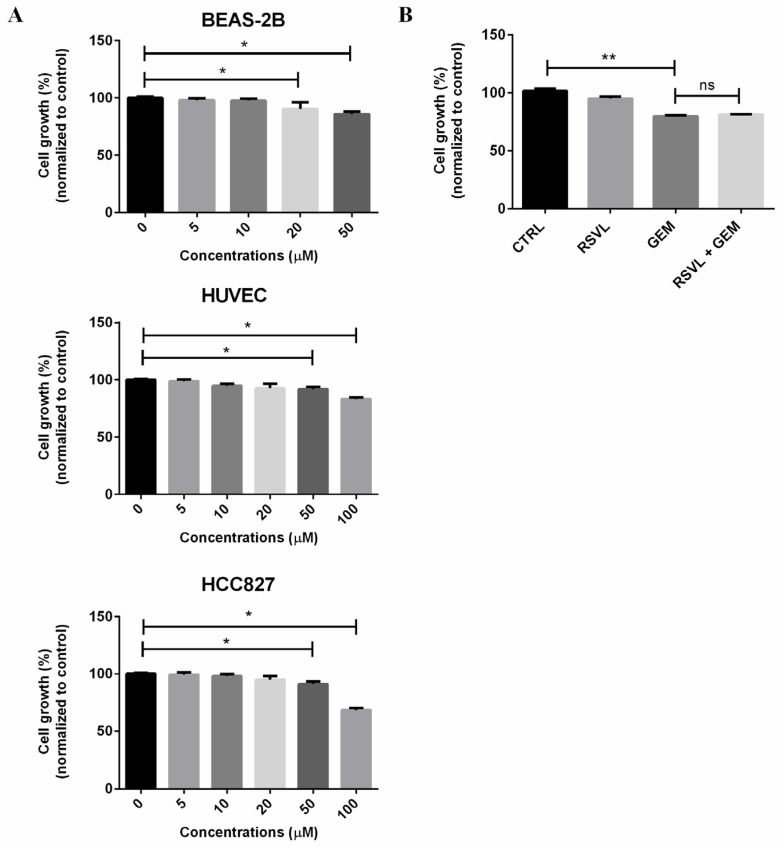
Resveratrol (trans-3,5,4′-trihydroxystilbene, RSVL) in combination with gemcitabine (GEM) showed no synergistic cytotoxic effects on HCC827 cancer cells cultured in vitro alone. (**A**) BEAS-2B, human umbilical vein endothelial cell (HUVEC), and HCC827 cells were treated with various concentrations of RSVL. (**B**) HCC827 cells were sham-exposed or treated with 10 μM RSVL and/or 1 μM GEM. After 24 h, the cell viability was measured by naphthol blue black (NBB) staining assay. The percentage of viability was plotted as 100% for control (no treatment of RSVL or GEM). Results are expressed as mean ± SD of triplicate samples, and reproducibility was confirmed in three separate experiments. * (*p* ≤ 0.05), ** (*p* ≤ 0.01), ns (not significant).

**Figure 2 cancers-12-00974-f002:**
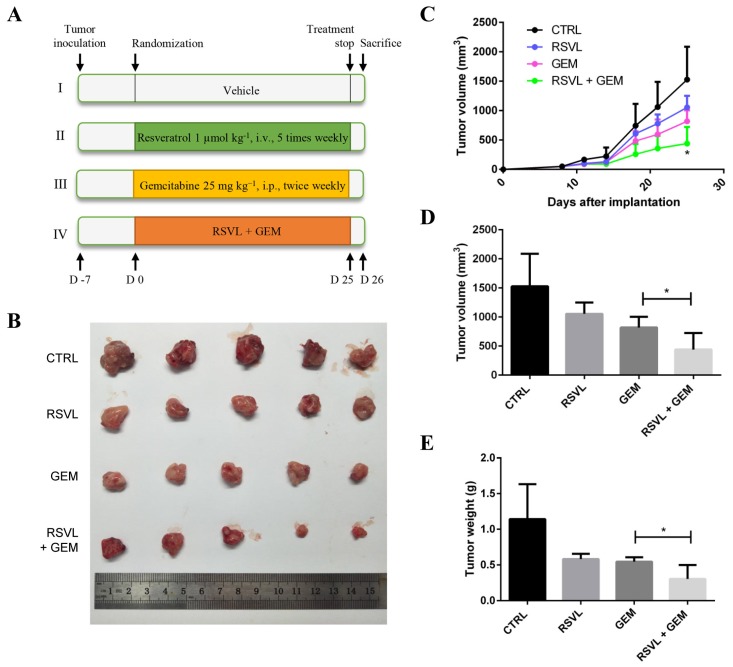
RSVL enhanced the anticancer efficacy of GEM in HCC827 lung cancer xenograft-implanted nude mice. (**A**) Schematic representation of the experimental protocol as described in the Materials and Methods section. A total of four mice groups were used. Group I was administrated with vehicle (100 μL, i.v. injection, five times weekly) and phosphate-buffered saline (100 μL, i.p. injection, twice weekly), group II was administrated with RSVL (1 µmol kg^−1^, i.v. injection, five times weekly), group III was administrated with GEM (25 mg kg^−1^, i.p. injection, twice weekly), and group IV was administrated with RSVL (1 µmol kg^−1^, five times weekly by i.v. injection) and GEM (25 mg kg^-1^, twice weekly by i.p. injection). (**B**) Image showing the excised tumor nodules from the above mice. (**C**) tumor volume measurement upon implantation of HCC827 cells in nude mice. (**D**) Comparison of tumor volumes at the last measurement. (**E**) Comparison of tumor weights at the last measurement. Values are mean ± SD and * (*p* ≤ 0.05) as compared with GEM-treated group alone.

**Figure 3 cancers-12-00974-f003:**
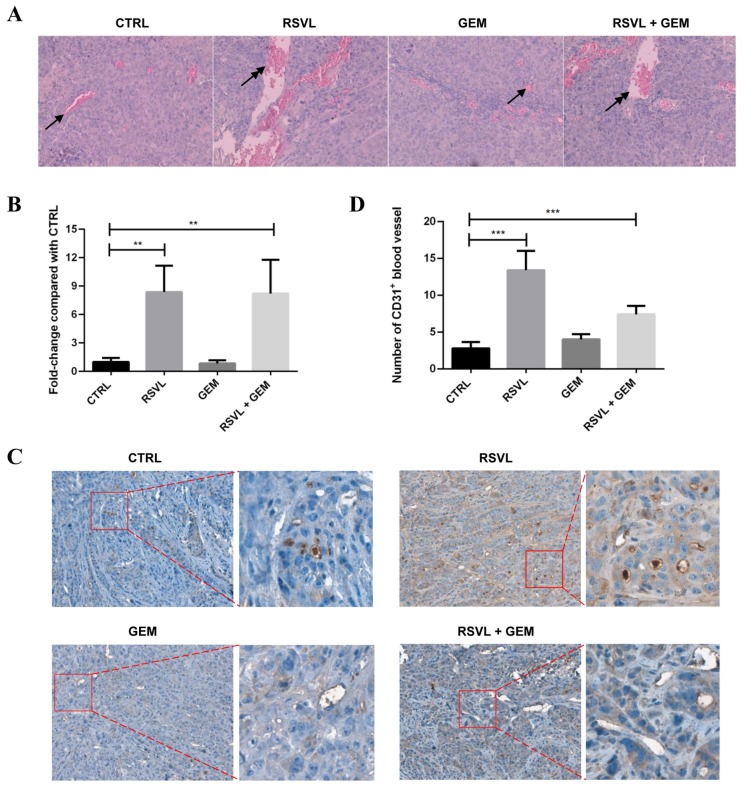
RSVL increased microvessel growth and promoted blood perfusion into tumor of lung cancer xenograft bearing nude mice. (**A**) Representative images of hematoxylin and eosin (HE) staining. Blood vessels formed in xenografts were indicated by arrows. Upon RSVL and RSVL + GEM treatment, more blood vessels, lacunae (indicated with double head arrows), as well as red-dyed blood cells can be seen. (**B**) The degree of blood perfusion in each group was quantitated and expressed as relative ratio, setting 1 for control. (**C**) Representative images of immunohistochemistry (IHC) staining for CD31 protein expression. Enlarged view is also shown on the right of each image. Upon RSVL and RSVL + GEM treatment, more CD31-positive staining blood vessels can be observed. (**D**) The number of CD31-positive staining blood vessels in each group was quantitated and presented. ** (*p* ≤ 0.01), *** (*p* ≤ 0.001).

**Figure 4 cancers-12-00974-f004:**
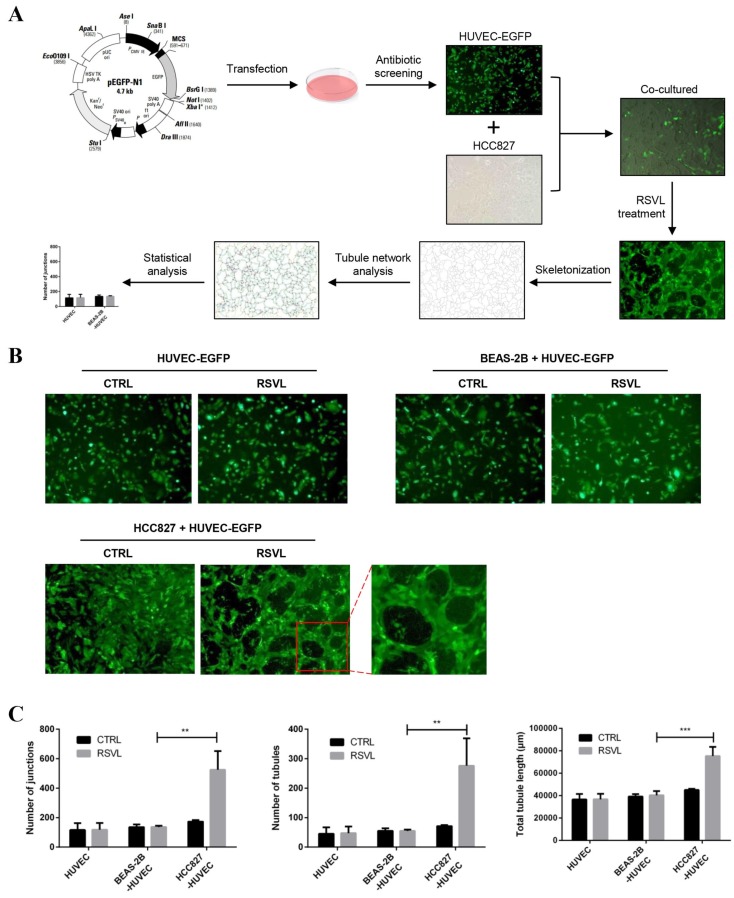
RSVL promoted tubule network formation in the HCC827-HUVEC co-culture model. (**A**) Schematic representation of experimental protocol as described in the Materials and Methods section. (**B**) Representative images taken under fluorescent microscope (10× magnification). HCC827-HUVEC appeared to have a more obvious tubule network formation than BEAS-2B-HUVEC or HUVEC alone. (**C**) The corresponding number of junctions, number of tubules, and total tubule length of the images of (**B**) were quantified and compared. ** (*p* ≤ 0.01), *** (*p* ≤ 0.001).

**Figure 5 cancers-12-00974-f005:**
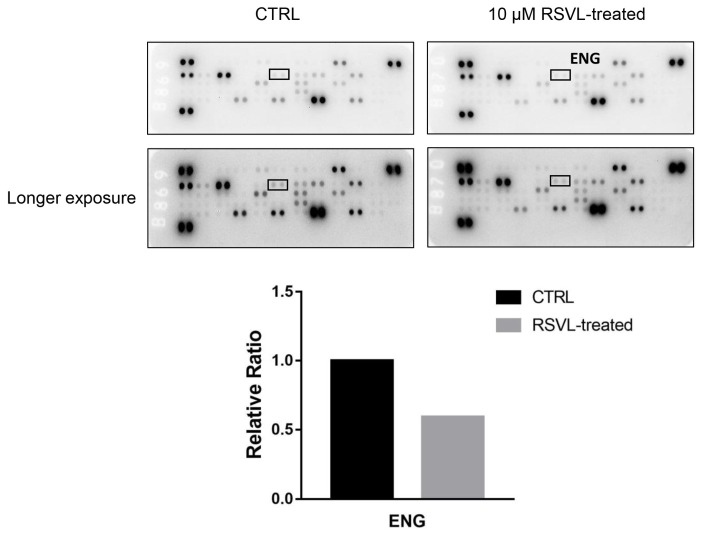
Human angiogenesis antibody array analysis identified endoglin (ENG) as one of the downregulated protein targets in 10 μM RSVL-treated HCC827. After 24 h of RSVL treatment, HCC827 cells were lysed and protein extract (300 μg) were used for angiogenesis array analysis. Array spots were visualized in accordance with the manufacturer’s instructions. The intensity of the spot was measured as described in the Materials and Methods section. The graph shows the relative ratios of ENG protein expression in cells, setting 1 for control (CTRL).

**Figure 6 cancers-12-00974-f006:**
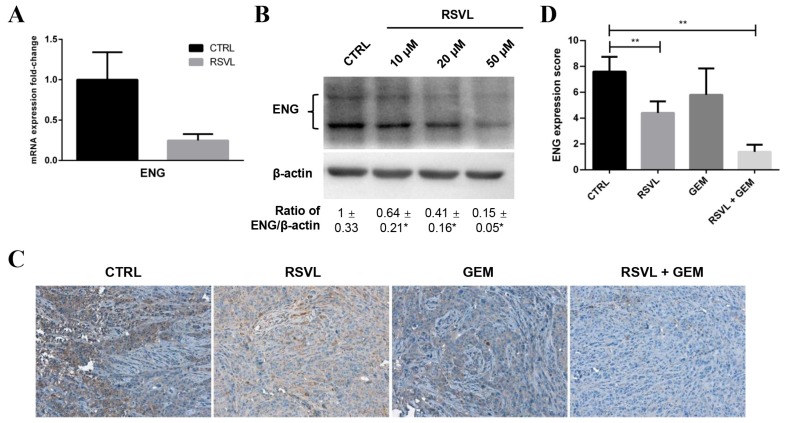
RSVL suppressed ENG expression both in vitro and in vivo. (**A**) Fold change of ENG mRNA expression in HCC827 cells cultured in vitro after treatment with 10 µM RSVL for 24 h. (**B**) RSVL suppressed the protein expression of ENG in a dose-dependent manner after treatment of HCC827 cells with RSVL from 10 to 50 µM for 24 h. (**C**) Representative images of IHC for ENG in tumor tissues from HCC827 xenograft nude mice. Positive ENG staining became weaker in RSVL group and in the combined treatment group as compared with the control or GEM alone group. (**D**) Comparsion for IHC score of ENG among each group in nude mice. IHC score of ENG decreased in RSVL group and in the combined treatment group as compared with control. * (*p* ≤ 0.05), ** (*p* ≤ 0.01).

**Figure 7 cancers-12-00974-f007:**
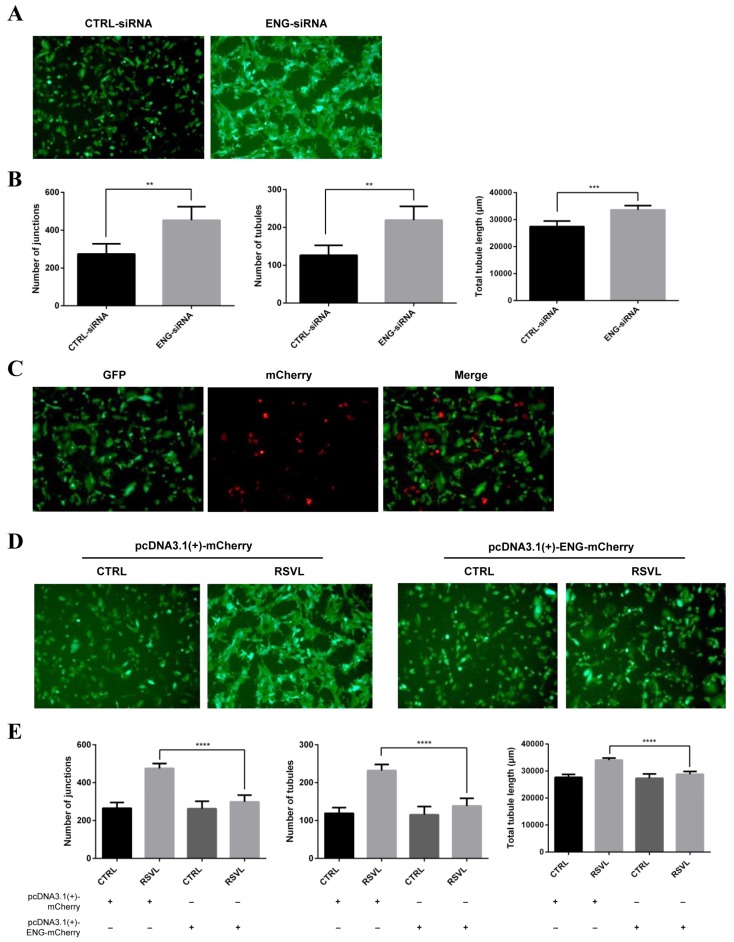
ENG is crucial in RSVL-promoted tubule network formation in HCC827-HUVEC co-culture model. (**A**) Representative images taken under the fluorescent microscope (10×) after knockdown of ENG. Upon transfection of HCC827 cells with ENG-siRNA for 24 h, HCC827 and HUVEC-EGFP cells were seeded at a ratio of 1:1 and co-cultured for 24 h, wherein tubule network formations were captured. HCC827-HUVEC showed better tubular formation, appearing to have a higher percentage of elongation, tubules, and junctions formed after ENG knockdown. (**B**) Quantification of tubule network formation in (**A**). Number of junctions (left), number of tubules (middle), and total tubule length (right). (**C**) Representative images of pcDNA3.1(+)-ENG-mCherry group taken under fluorescence microscope (10×). (**D**) Representative images after overexpression of ENG (10×). Upon transfection of HCC827 cells with pcDNA3.1(+)-mCherry or pcDNA3.1(+)-ENG-mCherry for 24 h, HCC827 and HUVEC-EGFP cells were seeded at a ratio of 1:1 and co-cultured for 24 h, wherein tubule network formations were observed and images of each group were captured. (**E**) Quantification of tubule network formation in (**D**). Number of junctions (left), number of tubules (middle), total tubule length (right). ** (*p* ≤ 0.01), *** (*p* ≤ 0.001), **** (*p* ≤ 0.0001).

**Figure 8 cancers-12-00974-f008:**
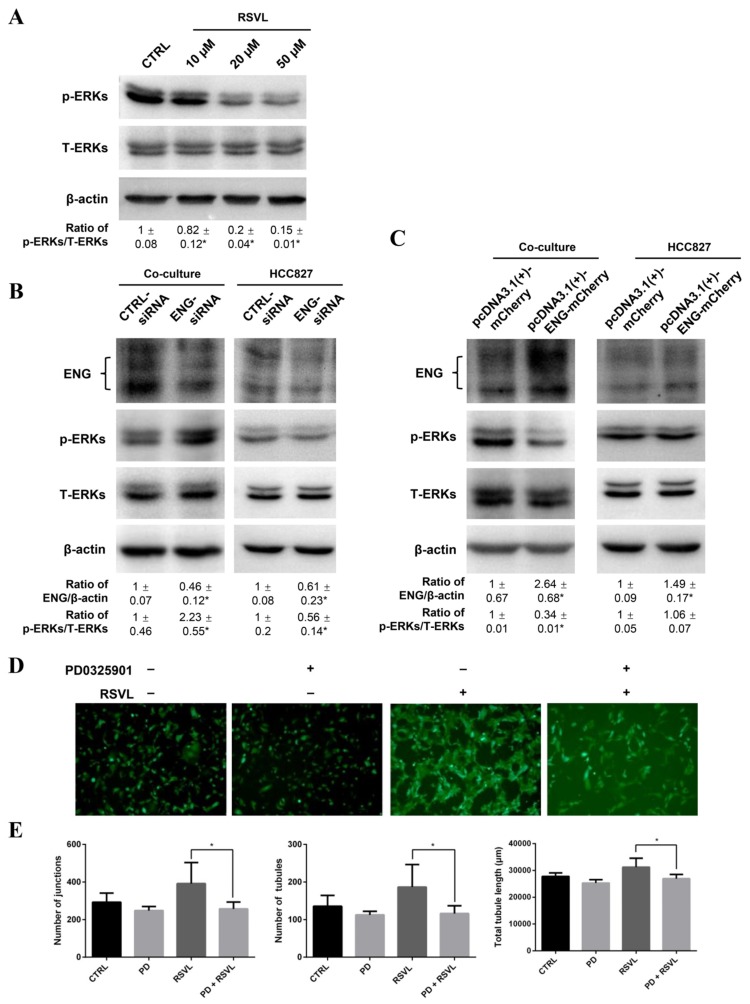
ENG and extracellular signal-regulated kinase (ERK) signaling pathway played important roles in RSVL-promoted tubule network formation in the HCC827-HUVEC co-culture model. (**A**) RSVL (10 to 50 µM for 24 h) inhibited the phosphorylation of ERKs in a dose-dependent manner in HCC827 cells. (**B**) Knockdown of ENG resulted in enhanced ERK1/2 activation in the HCC827-HUVEC co-culture model. (**C**) Overexpression of ENG resulted in suppressed ERK1/2 activation in the HCC827-HUVEC co-culture model. After transfected with ENG-siRNA (50 nM) or pcDNA3.1(+)-ENG-mCherry (3 µg) for 24 h in HCC827 cells, HCC827 and HUVEC-EGFP cells were seeded at a ratio of 1:1 to establish the co-culture model, and after 24 h of incubation, cell pellets were harvested and subjected to Western blot analyses. (**D**) Representative images taken under fluorescence microscope (10×). HCC827 and HUVEC-EGFP cells were seeded at a ratio of 1:1, and after 24 h of co-culture, cells were pretreated with PD0325901 for 1 h, then were treated with 10 µM RSVL for 24 h and tubule network formations were captured. HUVEC-EGFP showed better tubular formation after RSVL treatment, which can be abrogated by PD0325901 pretreatment. (**E**) Quantification of tubule network formation in (**D**). Number of junctions (left), number of tubules (middle), and total tubule length (right). * (*p* ≤ 0.05).

**Figure 9 cancers-12-00974-f009:**
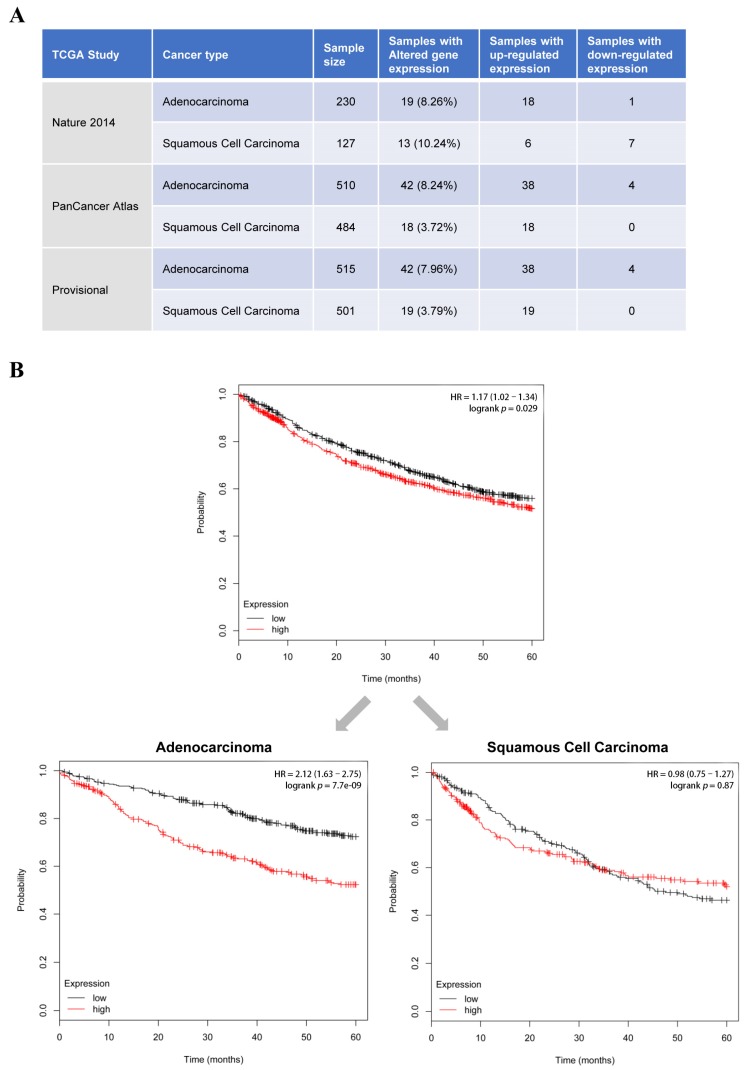
Altered expression of ENG was found in lung cancer patients. (**A**) Clinical datasets from The Cancer Genome Atlas (TCGA) were utilized to analyze the mRNA level of ENG in lung cancer patients with lung adenocarcinoma or lung squamous cell carcinoma. For each of the datasets, the mRNA expression z-score threshold was set at ± 1.5. (**B**) Kaplan–Meier plotter database was utilized to assess the correlation between ENG expression and survival of lung cancer patients.

**Figure 10 cancers-12-00974-f010:**
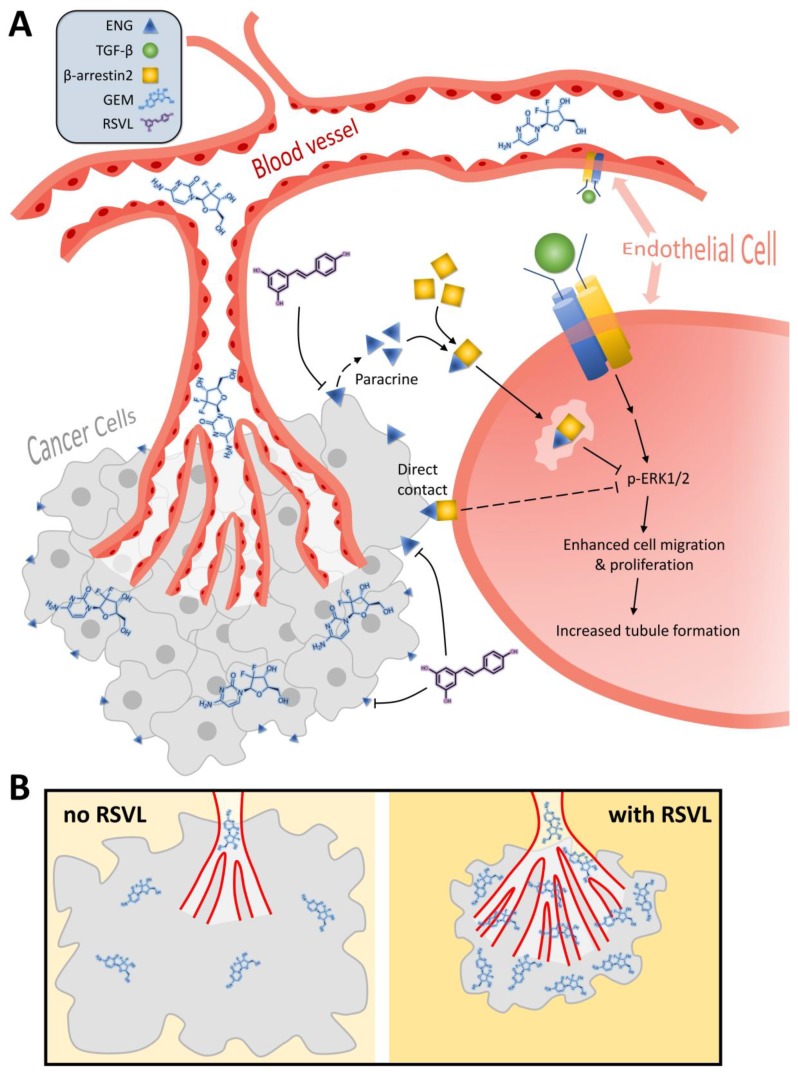
The putative molecular mechanism by which RSVL enhances tubule network formation and increases microvessel growth (**A**), leading to increased blood perfusion and drug delivery into tumor and thereby resulting in enhanced anticancer effect of GEM (**B**). The implications of our findings suggest the potential clinical applications of RSVL to enhance the anticancer efficacy of anticancer drugs against lung cancer.

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
