# Peer review of "Resveratrol Promotes Tumor Microvessel Growth via Endoglin and Extracellular Signal-Regulated Kinase Signaling Pathway and Enhances the Anticancer Efficacy of Gemcitabine against Lung Cancer"

_cancers, 2020, doi:10.3390/cancers12040974_

Round 1
Reviewer 1 Report
The authors showed that Resveratrol can enhances the anticancer efficacy of Gemcitabine in HCC827 lung cancer bearing nude mice in vivo but not in vitro. They also showed that Resveratrol Increases microvessel growth and promotes blood perfusion into tumor in lung cancer xenograft mice, promotes tubule network formation in a HCC827-HUVEC co-culture model. They concluded that Resveratrol can enhances the anticancer efficacy of Gemcitabine through suppresses the expression of ENG. But some of the results are not very clearly explained and hard to understand.
- In Figure 2B, the mouse body sizes are much smaller after Gemcitabine and Resveratrol/Gemcitabine combination treatment, the authors should normalize the tumor volume and tumor weight vs mouse body weight in Figure 1D-F.
- Figure 3, the authors showed the RSVL only also increased microvessel growth and promotes blood perfusion into tumor in lung cancer xenograft mice, but in Figure 2, they showed the RSVL alone did not change tumor growth. How did the authors explain the relationship between increased microvessel growth and blood perfusion into tumor and tumor growth?
- Figure 6, the author showed RSVL treatment alone also significantly reduced the ENG expression in mice compared with control, same as above, in Figure 2, RSVL alone did not change tumor growth. How did the authors explain the relationship between reduced ENG expression level by RSVL alone and tumor growth?
- In Figure 7, the authors need to provide the knockdown efficiency of siRNA targeting ENG. Same as overexpression of ENG, the authors need to provide the evidence that ENG was successfully over expressed at protein levels.
- All the western blots need to be quantified and a quantification graph with error bar needs to be provided.
- In Figure 10, since RSVL needs to work together with GEM to inhibit tumor growth, GEM needs to be included in the Figure 10.
Reviewer 2 Report
In this study authors showed that RSVL enhances the anticancer efficacy 474 of GEM against HCC827 lung cancer in vivo. Also they revealed a new molecular mechanism in which 475 ENG and ERK signaling pathway play important roles in RSVL-promoted tumor microvessel growth 476 and blood perfusion into tumor, which results in enhanced anticancer effect of GEM. Furthermore, the article is well constructed, the experiments were well conducted, and analysis was well performed.
I am accepting this article in the current format.
Reviewer 3 Report
The article “Resveratrol Promotes Tumor Microvessel Growth via Endoglin and Extracellular Signal-Regulated Kinase Signaling Pathway and Enhances the Anticancer Efficacy of Gemcitabine against Lung Cancer” by Quin et al. describes investigation of the mechanism behind the synergistic effect of gemcitabine and resveratrol in lung cancer. The study is well thought and clearly described. I believe that it could meet the interest of the readers of Cancers. Below I am listing some major and minor concerns to address by authors in order to improve the quality of their work.
Major corrections:
- Doubling time of cell lines selected for this study varies from 15 to 28h. It is recommended to perform cell viability assays to allow for at least two cell doubling times. The authors should explain why cell viability assay for all cell lines involved in the study was perform in the same time interval of 24h.
- The authors should provide an explanation on why Chou-Talalay method was not employed to determine synergistic drug concentrations.
- The authors should provide an explanation how resveratrol and gemcitabine doses and treatment schedule were determined for in vivo study.
- The authors should explain why cell viability assays (Figure 1A) were performed only for resveratrol alone, but not for gemcitabine alone. Also, please explain why gemcitabine was used in 1 uM concentration for combination study (Figure 1B).
Minor corrections:
- Line 115, please verify whether it should be petri dish or well, since the assay was performed in 24-well plate.
- Materials and Methods, please verify whether cell number per well is provided correctly, lines 393, 404, 414, shouldn’t it be 10^4, 10^5 and 10^6 respectively?
- Materials and Methods, Western Blotting Analysis should be described in more details to allow for experiment repetition. For example antibodies catalogue numbers and dilutions should be provided.
Round 2
Reviewer 1 Report
The revised manuscript was improved. The authors clarified the mouse body size is not different among groups and discussed the reviewers's questions why RSVL alone did not change tumor growth although changed microvessel growth and ENG expression. They also added quantification data of western blot band intensity. The quantification data need to be included in the same main figures instead of supplemental figures.
But the authors did not provide the evidence for knockdown efficiency of siRNA targeting ENG. siRNA may have off-target effect sometimes even knockdown efficiency was validated. If without knockdown efficiency validation, it is even harder to tell if there is any off-target effect. I would suggest the authors add the evidence to directly show the knockdown efficiency of siRNA again ENG.
Reviewer 3 Report
The authors addressed all the concerns.
Round 3
Reviewer 1 Report
The authors answered all the reviewer's questions. I think it is acceptable now.